

# The genomic sequence of *Exiguobacterium chiriqhucha* str. N139 reveals a species that thrives in cold waters and extreme environmental conditions

Ana Gutiérrez-Preciado[1,8,*], Carlos Vargas-Chávez[1,*], Mariana Reyes-Prieto[1], Omar F. Ordoñez[2], Diego Santos-García[1,9], Tania Rosas-Pérez[1], Jorge Valdivia-Anistro[3,4], Eria A. Rebollar[5], Andrés Saralegui[6], Andrés Moya[1], Enrique Merino[7], María Eugenia Farías[2], Amparo Latorre[1] and Valeria Souza[4]

[1] Unidad de Genética Evolutiva, Instituto Cavanilles de Biodiversidad y Biología Evolutiva, Universidad de Valencia, Calle Catedrático José Beltrán Martínez, Paterna, Valencia, Spain
[2] Laboratorio de Investigaciones Microbiológicas de Lagunas Andinas, Planta Piloto de Procesos Industriales Microbiológicos (PROIMI), Consejo Nacional de Investigaciones Científicas y Técnicas (CONICET), Av. Belgrano y Pasaje Caseros, San Miguel de Tucumán, Argentina
[3] Carrera de Biología, Faculta de Estudios Superiores Zaragoza, UNAM, Mexico City, Mexico
[4] Departamento de Ecología Evolutiva, Instituto de Ecología, Universidad Nacional Autónoma de México coyoacan, Mexico City, México
[5] Department of Biology, James Madison University, Harrisonburg, VI, United States of America
[6] Laboratorio Nacional de Microscopía Avanzada, Instituto de Biotecnología, Universidad Nacional Autónoma de México, Cuernavaca, Morelos, México
[7] Departamento de Microbiología Molecular, Instituto de Biotecnología, Universidad Nacional Autónoma de México, Cuernavaca, Morelos, México
[8] Current affiliation: Ecologie Systématique Evolution, CNRS, AgroParisTech, Université Paris Sud (Paris XI), Orsay, France
[9] Current affiliation: Department of Entomology, Hebrew University of Jerusalem, Rehovot, Israel
[*] These authors contributed equally to this work.

Corresponding authors
Amparo Latorre,
Amparo.Latorre@uv.es
Valeria Souza,
souza.valeria2@gmail.com

## ABSTRACT

We report the genome sequence of *Exiguobacterium chiriqhucha* str. N139, isolated from a high-altitude Andean lake. Comparative genomic analyses of the *Exiguobacterium* genomes available suggest that our strain belongs to the same species as the previously reported *E. pavilionensis* str. RW-2 and *Exiguobacterium* str. GIC 31. We describe this species and propose the *chiriqhucha* name to group them. 'Chiri qhucha' in Quechua means 'cold lake', which is a common origin of these three cosmopolitan Exiguobacteria. The 2,952,588-bp *E. chiriqhucha* str. N139 genome contains one chromosome and three megaplasmids. The genome analysis of the Andean strain suggests the presence of enzymes that confer *E. chiriqhucha* str. N139 the ability to grow under multiple environmental extreme conditions, including high concentrations of different metals, high ultraviolet B radiation, scavenging for phosphorous and coping with high salinity. Moreover, the regulation of its tryptophan biosynthesis suggests that novel pathways remain to be discovered, and that these pathways might be fundamental in the amino acid metabolism of the microbial community from Laguna Negra, Argentina.

## INTRODUCTION

The high altitude Andean Lakes (HAALs) from Puna, Argentina, are a group of lakes located at 3,000–6,000 m above sea level which are characterized by high ultraviolet (UV) radiation and salinity, broad temperature variations, low nutrient concentrations and high contents of metals and metalloids, mainly arsenic (*Fernández-Zenoff et al., 2006*; *Fernández-Zenoff, Siñeriz & Farías, 2006*; *Dib et al., 2008*; *Flores et al., 2009*; *Ordoñez et al., 2009*; *Albarracín et al., 2011*; *Belfiore, Ordoñez & Farías, 2013*). These environmental conditions are considered to be extreme and might resemble those of the Earth's early atmosphere, as has been stated by NASA (*Cabrol et al., 2007*; *Farías et al., 2009*). Hence, these geographical areas have been proposed for studies on astrobiology (*Farías et al., 2009*). Despite being oligotrophic and hostile, a great microbial diversity has been found in the HAALs, where bacteria from the genus *Exiguobacterium* are one of the dominant taxa (*Ordoñez et al., 2009*; *Ordoñez et al., 2013*; *Sacheti et al., 2013*).

The *Exiguobacterium* genus, a sister clade to the Bacillus genus, is currently underexplored, and molecular studies of this genus from different sources are limited (*Vishnivetskaya, Kathariou & Tiedje, 2009*). Exploring *Exiguobacterium* strains is of great significance because understanding their strategies to adapt to diverse and extreme environmental conditions will likely place them as model organisms involved in the remediation of organic and inorganic pollutants. In particular, *Exiguobacterium* strains isolated from the HAALs have the potential of becoming an attractive model system to study environmental stress responses, as these microorganisms are able to grow efficiently in the laboratory (*Ordoñez et al., 2009*; *Belfiore, Ordoñez & Farías, 2013*). Moreover, *Dib et al. (2008)* suggested that these microorganisms could harbor various stress defense associated systems.

The *Exiguobacterium chiriqhucha* str. N139 was selected for genome sequencing due to its stress defense mechanisms such as its tolerance to high UV-B radiation, salinity and metalloids, particularly arsenic. This strain was isolated from the water column of Laguna Negra, which belongs to the 'Salar de la Laguna Verde', a system of five shallow oligotrophic lakes originated in the Tertiary (65 million to 1.8 million years ago) (*Ericksen & Salas, 1987*).

In the present study we characterized the genome of *E. chiriqhucha* str. N139, in order to identify the strategies that this organism employs to cope with the extreme environmental factors present in the aforementioned lake, mainly those related to metal and UV-B resistance. We also performed comparative genomics focusing on the two strains that would comprise the same species as *Exiguobacterium chirqhucha* str. N139; *E. pavilionensis* str. RW-2, isolated from the permanently cold Pavilion Lake in Canada (*White III, Grassa & Suttle, 2013*) and *Exiguobacterium* str. GIC31 isolated from a glacier in Greenland (*Vishnivetskaya et al., 2014*). Since the three strains were isolated from cold lakes, we propose the name 'chiri qhucha', which means 'cold lake' in Quechua, the Andean prehispanic language.
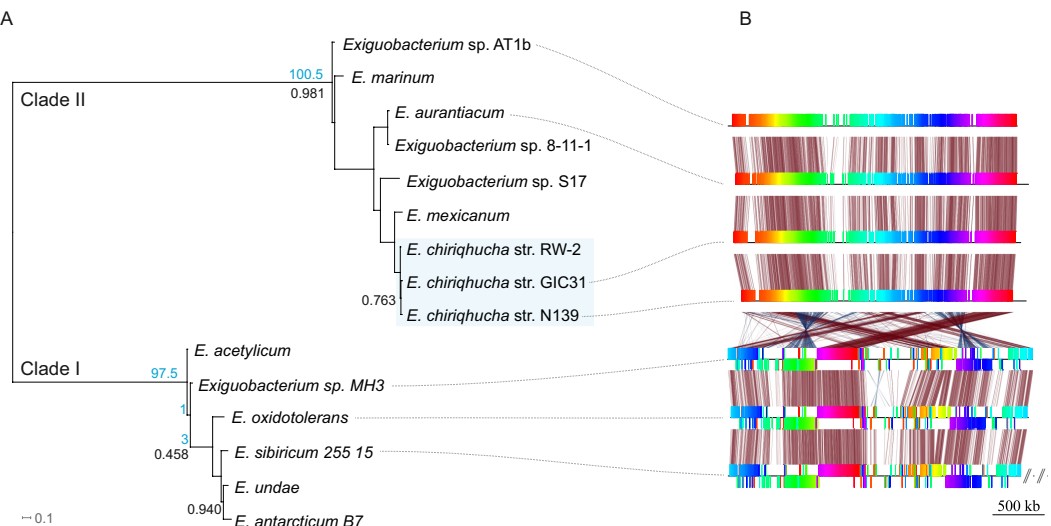

**Figure 1** **Evolutionary history of the genus *Exiguobacterium*.** (A) Phylogenetic reconstruction using complete genomic sequences of 17 representative *Exiguobacterium* strains. The tree was built with Phy-loPhlAn (*Segata et al., 2013*). (B) Synteny among *Exiguobacterium* strains. Nucleotide syntenic blocks are represented by colored bars. Red links denote no rearrangements between the blocks compared. Blue links denote rearrangements between the blocks compared. Blue numbers in the phylogeny denote the minimum number of rearrangements obtained with *MGR*. Plasmids from *E. sibiricum* are displayed at the right (separated from the chromosome by backslashes). Black numbers indicate bootstrap values different from 100%.

## Classification and features

Members of the *Exiguobacterium* genus are Firmicutes, Gram-positive, facultative anaerobes with a low G + C content (*Vishnivetskaya, Kathariou & Tiedje, 2009*). *Exiguobacterium* is widely distributed all over the world (*Karami et al., 2011*) and has been isolated and typified from a wide variety of environments including hot springs (*Vishnivetskaya, Kathariou & Tiedje, 2009*; *Vishnivetskaya et al., 2011*), hydrothermal vents (*Crapart et al., 2007*), permafrost (*Vishnivetskaya & Kathariou, 2005*; *Vishnivetskaya et al., 2006*; *Rodrigues et al., 2008*), marine sediment (*Kim et al., 2005*), oligotrophic environments (*Rebollar et al., 2012*), biofilms (*Carneiro et al., 2012*), alkaline methanogenic microcosms (*Rout, Rai & Humphreys, 2015*) and more recently in water and microbial mats from high-altitude desert wetlands (*Ordoñez et al., 2013*). The *Exiguobacterium* genus is divided in two main phylogenetic clades (*Vishnivetskaya, Kathariou & Tiedje, 2009*); clade I is composed of temperate and cold-adapted strains, whereas clade II includes alkaliphilic species, with a marine origin and/or from high-temperature habitats (Fig. 1A).

*E. chiriqhucha* str. N139, which belongs to clade II, was isolated from the water column of Laguna Negra, in the HAALs (GPS: 27°38′49″S, 68° 32′43″W) and in laboratory conditions can uptake a wide variety of carbon sources (Table S1). Its cells are short rods and do not sporulate (Fig. 2, Table 1).

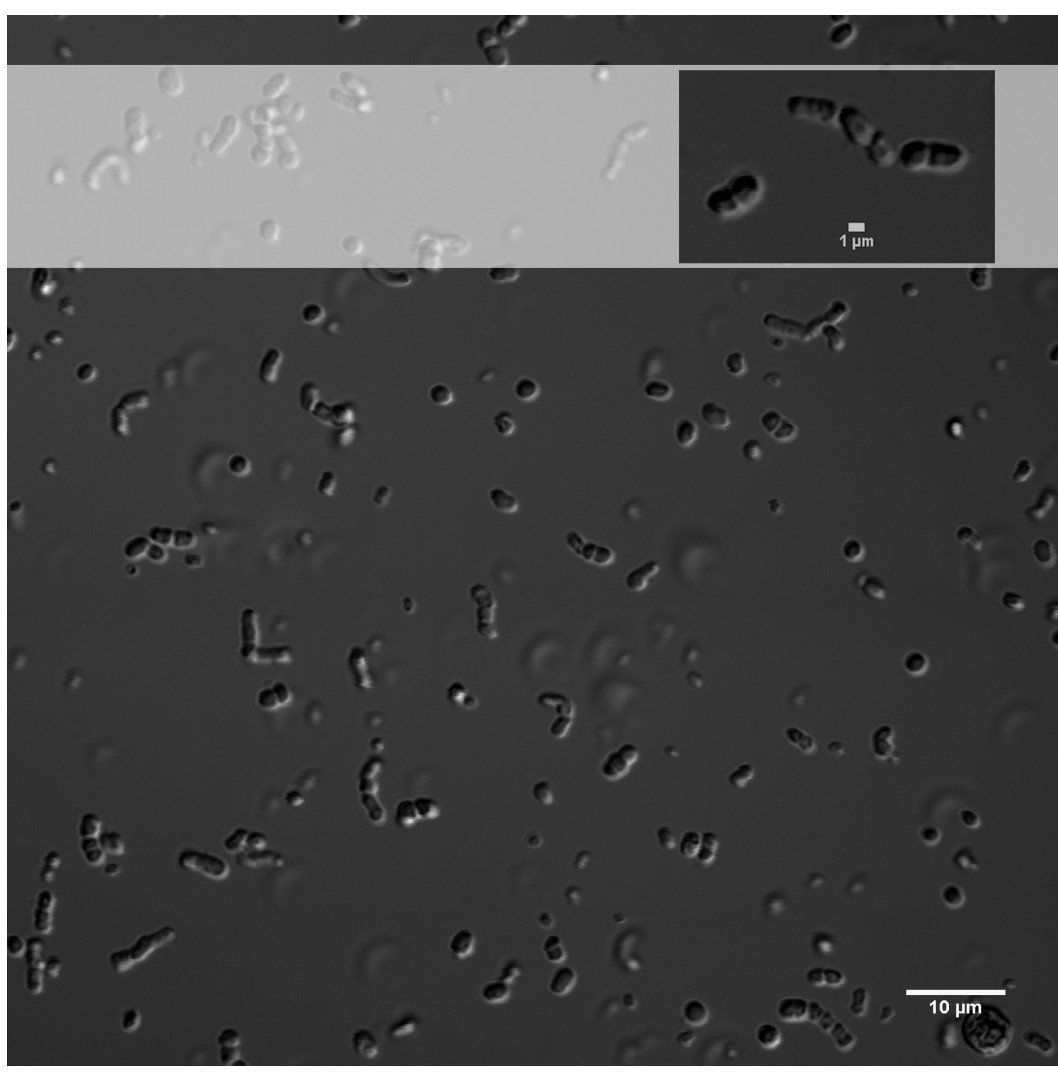

**Figure 2   Differential interference contrast image of *E. chiriqhucha* str. N139.**

## MATERIALS AND METHODS

### Growth conditions and genomic DNA preparation

*E. chiriqhucha* str. N139 was isolated from Laguna Negra by plating it in Lake Medium (LM). LM was used to maintain the same salinity as the isolation environment and was obtained by filtering lake water (0.22 μm Biopore filters) and adding 2.5 g of yeast extract and 12 g of agar (Difco) per liter at 20 °C. For future assays the strain was grown in LM broth at 20 °C with agitation. DNA was extracted using the protocol described by *Fernández-Zenoff, Siñeriz & Farías (2006)*.

### Microscopy

Differential interference contrast (DIC) images were obtained from cells grown on LB medium overnight, and mounted in No. 2 coverslips (Fig. 2). LB medium was used as mounting media during image acquisition. Images were shot with an Olympus FV1000

**Table 1  Classification and general features of *Exiguobacterium chiriqhucha* str. N139.**

| Property | Term | Evidence code[a] |
|---|---|---|
| Classification | Domain *Bacteria* | TAS (*Woese, Kandler & Wheelis, 1990*) |
| | Phylum Firmicutes | TAS (*Gibbons & Murray, 1978*) |
| | Class Bacilli | TAS (*De Vos et al., 2009*) |
| | Order Bacillales | TAS (*De Vos et al., 2009*) |
| | Family Bacillales *Family XII. Incertae Sedis* | TAS (*De Vos et al., 2009*) |
| | Genus *Exiguobacterium* | TAS (*De Vos et al., 2009*; *Vishnivetskaya, Kathariou & Tiedje, 2009*) |
| | Species *Exiguobacterium chiriqhucha* | TAS (*White III, Grassa & Suttle, 2013*) |
| | Strain: *N139 (Accession: JMEH00000000)* | |
| Gram stain | *Positive* | IDA |
| Cell shape | *Short rods* | IDA |
| Motility | *Motile* | IDA |
| Sporulation | *Non-sporulating* | EXP |
| Temperature range | *Mesophilic (30–37 °C)* | IDA |
| Optimum temperature | *30 °C* | IDA |
| pH range; Optimum | *7–9* | IDA |
| Carbon source | *β—Methylglucoside, Galacturonic acid, L-asparagine, Tween 40, L-Serine, N-acetyl-glucosamine, Hydroxybutyric acid, Itaconic acid, Ketobutyric acid, Putrescine (See Table S1)* | EXP |
| Habitat | *Aquatic* | TAS (*Flores et al., 2009*) |
| Salinity | *0.11%–10% NaCl (w/v)* | IDA |
| Oxygen requirement | *Facultatively anaerobic* | TAS (*De Vos et al., 2009*) |
| Biotic relationship | *free-living* | IDA |
| Pathogenicity | *non-pathogen* | NAS |
| Geographic location | *Laguna Negra, Catamarca, Argentina* | IDA |
| Sample collection | *2006* | IDA |
| Latitude | *27° 39′ 20.17″ S* | IDA |
| Longitude | *68° 33′ 46.18″ W* | IDA |
| Altitude | *4100 masl* | IDA |

**Notes.**

[a]Evidence codes—IDA, Inferred from Direct Assay; TAS, Traceable Author Statement (i.e., a direct report exists in the literature); NAS, Non-traceable Author Statement (i.e., not directly observed for the living, isolated sample, but based on a generally accepted property for the species, or anecdotal evidence); EXP, Inferred from Experiment. These evidence codes are from the Gene Ontology project (*Gene Ontology Evidence Codes*).

Laser Scanning Confocal on an Olympus IX81 inverted microscope equipped with 60x UPlanSApo NA 1.3 Sil objective lens. With a 405nm laser line, DIC Images were acquired in the TD channel controlled with Olympus FV10-ASW-4.2 software. Brightness, contrast and scale bars were adjusted on displayed images using the Fiji software.

## Phylogenetic reconstruction

The complete genomic sequences of 17 representative *Exiguobacterium* strains were used to reconstruct their phylogeny with PhyloPhlAn. This software extracts a set 31 manually curated conserved proteins from each genome, aligns them, keeps the positions that retain important evolutionary information and builds a phylogenetic tree (*Segata et al., 2013*).

**Table 2 Project information.**

| Property | Term |
| --- | --- |
| Finishing quality | Permanent-draft |
| Libraries used | 454 pyrosequence standard library |
| Sequencing platforms | 454 Titanium |
| Fold coverage | 85× |
| Assemblers | Newbler 2.8 and MIRA 3.4 |
| Gene calling method | Prokka |
| Locus Tag | EF88 |
| Genbank ID | JMEH00000000.1 |
| GenBank Date of Release | December, 2015 |
| GOLD ID | Go0093977 |
| BIOPROJECT | PRJNA245187 |
| Source Material Identifier | N139 |
| Project Relevance | UV resistance, metal resistance, adaptation to oligotrophic environments |

Figure 1A shows this phylogenetic reconstruction of the selected *Exiguobacterium* genomes from both clades.

## Genome sequencing and assembly

The genome of *E. chiriqhucha* str. N139 was generated using *454 technology* (Table 2). A standard *454 Titanium* library was constructed and sequenced, producing 664,086 reads, totaling 253.9 Mb of data. Phred quality cut-off was set to 20. The 454 data was assembled with *Newbler*, version 2.8 and MIRA, version 3.4 (*Chevreux et al., 2004*). The GS De Novo Assembler GUI was used, parameters selected were minimum read length of 45 and output scaffolds file. The parameters used for the MIRA assembly were: number of passes of 1 and no uniform read distribution nor trimming of overhanging reads. Warning message for read names longer than 40 was deactivated. Both assemblies were merged using *Minimus2*, from the *amos* version 3.1.0, with assembly errors manually corrected. The contigs were sorted with *Mauve* version 2.3.1 (*Rissman et al., 2009*), using *Exiguobacterium* sp. AT1b as the reference because it is the closest relative with a completely sequenced genome (*Vishnivetskaya et al., 2011*).

## Genome annotation

Protein-coding genes, tRNAs, rRNAs and non-coding RNAs were identified using the annotation pipeline *Prokka* (*Seemann, 2014*), followed by annotation refinement with *Inter-ProScan* (*Quevillon et al., 2005*). Riboswitches were identified with the *Infernal* 1.1 package (*Nawrocki & Eddy, 2013*) using the corresponding covariance models from the *Rfam* database (*Burge et al., 2013*). COGs were assigned by profile hidden Markov model (profile HMM) searches using the *hmmsearch* program of the *HMMER3* package (*Mistry et al., 2013*). For every COG, a multiple sequence alignment of *bona fide* representative sequences were generated using the *Muscle* program (*Edgar, 2004*); then, the corresponding Hidden Markov Model was built using the *hmmerbuild* program, also provided in the

*HMMER3* package (*Mistry et al., 2013*). The cutoff *E*-value in the *hmmsearch* process varies importantly for every COG. For every one of the COG groups we have defined a high confidence cutoff *E*-value value defined as the highest *E*-value (smallest bit score) observed for the members of such COG. In any case, none of the COG cutoff *E*- values was greater than 1e–10. The resulting annotation was subjected to manual curation. *Pathway Tools 13 (Karp et al., 2009)* in combination with the *BioCyc* (*Caspi et al., 2014*) and *UNIPROT* (*UniProt Consortium, 2015*) databases were used to infer the metabolic capacities of *E. chiriqhucha* str. N139. The curated model of *E. chiriqhucha* str. N139 can be provided upon request, and will be deposited in the *BioCyc* database.

## CLUSTERING OF STRAINS INTO A SINGLE SPECIES

Phylogenetic analyses, ANI and AAI calculations, synteny analyses, as well as pangenome reconstruction were performed in order to a better understanding of the taxonomic position of the strain N139 isolated from Laguna Negra, relative to all the *Exiguobacterium* genomes available at the time of analysis.

ANI and AAI calculations (*Goris et al., 2007*) were done with default parameters for N139 *versus* all other complete genomic sequences of *Exiguobacterium* as well as pairwise comparisons for the three closest strains using the web server from Kostas lab http://enve-omics.ce.gatech.edu/ with default parameters. Comparisons to *Exiguobacterium* str. N139 can be seen in Table S2.

To explore the genomic rearrangements present on *E. chiriqhucha* str. N139 in comparison to other *Exiguobacterium* species, nucleotide syntenic blocks were obtained with *Mauve* version 2.3.1 (*Darling et al., 2004*). Syntenic block permutations were exported and used as input for *MGR* (*Bourque & Pevzner, 2002*). *MGR* was used to calculate the minimum number of rearrangements between the species analyzed, and to recover the rearrangement dendrogram. *genoPlotR* (*Guy, Kultima & Andersson, 2010*) was used to plot the syntenic blocks (Fig. 1B).

The pangenome of the nine *Exiguobacterium* genomes from clade II available at the time was reconstructed to aid in the taxonomic positioning of the strain N139. Orthologs were first calculated following the OrthoMCL pipeline (*Li, Jr & Roos, 2003*; *Fischer et al., 2011*), and the pangenome and the core genome were elucidated using *ad hoc* perl scripts. The selected strains were *E.* str. AT1b isolated from a hot spring in Yellowstone, USA (*Vishnivetskaya et al., 2011*); *E. marinum*, isolated from the Yellow Sea, South Korea (*Vishnivetskaya et al., 2014*); *E. aurantiacum*, from a potato processing plant, in the UK (*Vishnivetskaya et al., 2014*); *E.* str. 8-11-1 isolated from a salt lake in Inner Mongolia, China (*Jiang et al., 2013*); *E.* sp. S17 from the Laguna Socompa, another HAAL, Argentina (*Ordoñez et al., 2013*); *E. mexicanum* isolated from a brine shrimp *Artemia franciscana* (*López-Cortés et al., 2006*); *E. pavilionensis* (now *chiriqhucha*) str. RW-2 isolated from Pavilion Lake, Canada (*White III, Grassa & Suttle, 2013*) and *E.* (now *chiriqhucha*) str. GIC31, isolated from glacier ice in Greenland (*Vishnivetskaya et al., 2014*).

## UV RESISTANCE ASSAYS: DETERMINATION OF SURVIVAL RATE

The strains *Exiguobacterium* sp. S17, isolated from Laguna Socompa (HAAL), and *Exiguobacterium aurantiacum* str. DSM 6208, from the German Collection of Microorganisms and Cell Cultures (DSM), were used in these assays for comparison as external controls. These strains and *E. chiriqhucha* str. N139 were grown in 40 mL of LB medium under shaking (150 rpm) at 30 °C and cells were harvested by mid log phase (OD$_{600 \text{ nm}}$ 0.5) by centrifugation at 8,000 rpm for 10 min at 4 °C. The pellets were washed twice with 30 mL of 0.9% NaCl, and resuspended in the same volume of 40 mL. 20 mL of cell suspensions were transferred into sterile quartz tubes (16 cm long and 1.8 cm diameter) and placed horizontally to ensure maximal exposure and incubated at 15 °C under gentle shaking (150 rpm).

Tubes were irradiated from a distance of 30 cm with UV-B doses between 2,0 - 3,0 W/m2 during 240 min (09815-06 lamps, Cole Parmer Instrument Company; major emission line at 312 nm). Tubes were covered with an acetate sheet to block out UV-C. Irradiance was quantified with a radiometer (09811-56, Cole Parmer Instrument Company) at 312 nm with half bandwidth of 300 to 325 nm. Aliquots of 0.1 mL were taken at different exposure times (0, 60, 120, 180 and 240 min). Samples were then serially diluted in LB broth and spread in duplicate on Petri dishes with the same medium to determine the number of colony forming units (CFU). Controls of unexposed samples were run simultaneously in darkness and the percentage of cell survival after each treatment was calculated relative to these controls.

## RESULTS AND DISCUSSION

### Genome properties

The final assembly of the genome of *E. chiriqhucha* str. N139 consists of 23 contigs, the smallest one being 457 bases in length and the largest 1.5 Mb, with an average coverage of 85×. Its genome includes three circular megaplasmids with probable sizes of 250.57, 137.48 and 48 Kb, as determined by Pulse Field Gel Electrophoresis (PFGE) analysis (see Fig. S1 and Supplemental Information 1) and one circular chromosome with an estimated size of 2,516 kb, with a 52% GC content. A total of 3,182 genes were predicted (3,049 protein-coding genes and 82 noncoding RNA genes (95.8% and 2.57% respectively)). *E. chiriqhucha* str. N139 has 10 ribosomal rRNA operons, confirmed by PFGE (see Supplemental Information 1 and Fig. S2). A putative function was assigned to 2,214 (73%) of the protein-coding genes, and the remaining genes were annotated as hypothetical proteins. The properties and the statistics of the genome are summarized in Table 3. 2,575 protein-coding genes were assigned to 1,603 COG families, corresponding to a gene content redundancy of 38.1% (see Table 4).

### Genome rearrangements

Genome rearrangements within clades I and II are scarce, showing high conservation of the genomic structure within clades. However, several genomic rearrangements occurred as both clades diverged.

**Table 3  Nucleotide content and gene count levels of the *E. chiriqhucha* str. N139 genome.**

| Attribute | Genome (total) | |
| --- | --- | --- |
| | Value | % of total[a] |
| Genome size (bp) | 2,952,588 | – |
| DNA coding (bp) | 2,655,834 | 89.94 |
| DNA G + C (bp) | | 52 |
| DNA Scaffolds | 23 | |
| N50 | 1,553,709 | |
| Total genes | 3,182 | 100 |
| RNA genes | 82 | 2.62 |
| Protein-coding genes | 3,049 | 95.82 |
| Pseudogenes | 26 | 0.81 |
| Genes in internal clusters | NA | |
| Genes with function prediction | 2,356 | 74.04 |
| Genes assigned to COGs | 2,575 | 80.92 |
| Genes with Pfam domains | 2,538 | 79.76 |
| Genes with signal peptides | NA | |
| Genes with transmembran helices | 888 | 27.90 |
| CRISPR repeats | 0 | |

**Notes.**

[a]The total is based on either the size of the genome in base pairs or the total number of protein coding genes in the annotated genome.

In order to determine which contigs of *E. chiriqhucha* str. N139 belong to plasmids, the plasmid sequences of pEspA and pEspB from *E. arabatum* RFL1109 (*Jakubauskas et al., 2009*) were retrieved from NCBI. This strain was selected for comparison because their plasmids have been widely studied (*Jakubauskas et al., 2009*) and because it is phylogenetically close to *E. chiriqhucha* str. N139. Jakubauskas and colleagues identified the regions hr-A1, hr-AB and hr-A2 in plasmid pEspA as capable of replicating the plasmid in *Bacillus*. For plasmid pEspB, they hypothesized that the regions hr-B1, hr-AB and hr-B2 are involved in a theta replication mechanism (*Jakubauskas et al., 2009*). BLAST searches of these regions were performed against all *Exiguobacterium* genome sequences available to date. For the strains *E.* MH3, *E. antarcticum* and *E.* sp. AT1b, which are described as genomes without plasmids, no significant hits were found. Conversely, hits to the *E. arabatum* sequences hr-B1, hr-AB and hr-B2 (a fragment of 39 kb) were found in the genomes of *E. chiriqhucha* str. GIC31 (56 kb) and *E. chiriqhucha* str. N139 (contig000014 of size 25 kb). It was concluded that the sequences present in the plasmids are shared within different *Exiguobacterium* strains, displaying a highly dynamic behavior. Therefore it was not possible to determine which of our contigs correspond to the three megaplasmids observed in the PFGE experiments (see Supplemental Information 1). Furthermore, contig 14 in our assembly corresponds to the smallest contig of *E. chiriqhucha* str. GIC31, so it could be a plasmid in both *Exiguobacterium* strains. Genes belonging to contig 14 are mostly hypothetical proteins, only 11 genes could be annotated. Of these, 9 correspond to genes involved in mobile elements (antirestriction proteins, integrases and transposases),

**Table 4 Genes associated with the 25 general COG functional categories.**

| Code | Value | % of total[a] | Description |
|---|---|---|---|
| J | 166 | 5.44 | Translation, ribosomal structure and biogenesis |
| A | 0 | 0 | RNA processing and modification |
| K | 235 | 7.71 | Transcription |
| L | 144 | 4.72 | Replication, recombination and repair |
| B | 1 | 0.03 | Chromatin structure and dynamics |
| D | 36 | 1.18 | Cell cycle control, Cell division, chromosome partitioning |
| V | 62 | 2.03 | Defense mechanisms |
| T | 166 | 5.44 | Signal transduction mechanisms |
| M | 144 | 4.72 | Cell wall/membrane biogenesis |
| N | 75 | 2.46 | Cell motility |
| U | 53 | 1.74 | Intracellular trafficking and secretion |
| O | 100 | 3.28 | Posttranslational modification, protein turnover, chaperones |
| C | 152 | 4.99 | Energy production and conversion |
| G | 232 | 7.61 | Carbohydrate transport and metabolism |
| E | 224 | 7.35 | Amino acid transport and metabolism |
| F | 84 | 2.76 | Nucleotide transport and metabolism |
| H | 97 | 3.18 | Coenzyme transport and metabolism |
| I | 81 | 2.66 | Lipid transport and metabolism |
| P | 170 | 5.58 | Inorganic ion transport and metabolism |
| Q | 54 | 1.77 | Secondary metabolites biosynthesis, transport and catabolism |
| R | 463 | 15.19 | General function prediction only |
| S | 327 | 10.72 | Function unknown |
| – | 447 | 15.55 | Not in COG |

**Notes.**
[a] The total is based on the total number of protein coding genes in the annotated genome.

conjugal transfer proteins and competence factors; one antibiotic resistance gene and a RNaseH. However, contig 14 lies adjacent to contig 13, both accounting for a total size of 100 kb when synteny was evaluated against *E. chiriqhucha* (*pavillionensis*) str. RW-2. It is worth mentioning that contig 13 possesses most of the genes responsible for metals resistance, but this region appears to be integrated in the chromosome of *E. chiriqhucha* str. GIC31. This highly dynamic behavior across strains, along with the presence of several genes involved in mobility, suggests that, if both contigs belong to a plasmid, it might be an integrative one. A MAUVE analysis performed between the three *E. chiriqhucha* strains, N139, GIC31 and RW-2, shows high synteny across their chromosomes. This idea that contigs 12, 13 and 14 might belong to the plasmids is supported by their shifts in GC skew (Fig. 3). To all appearances, the chromosomes within each of the two main clades of the *Exiguobacterium* species are very similar, but quite distinct when compared between these clades (Fig. 1).

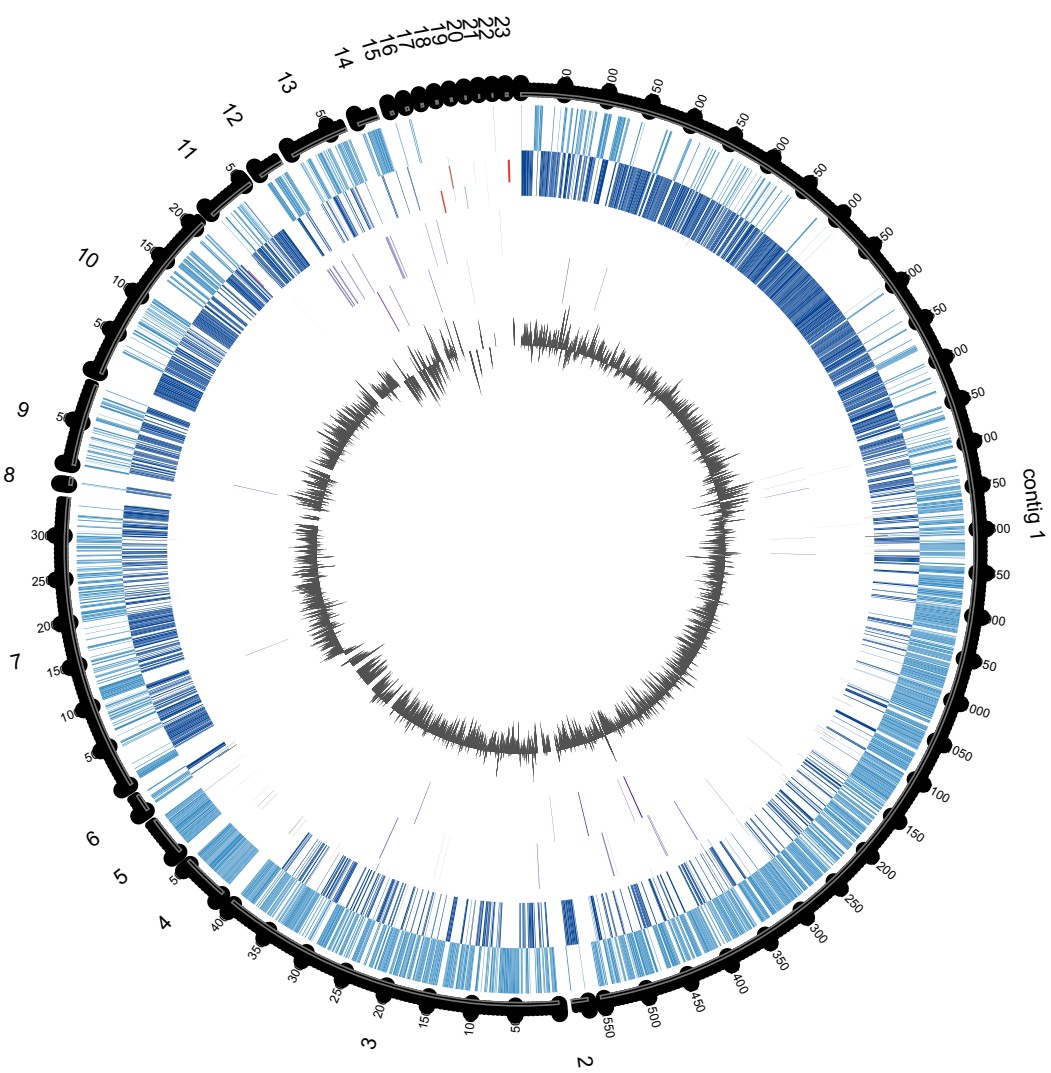

**Figure 3** **Circular genome map of *E. chiriqhucha* str. N139.** Circle tracks from out towards inside are as follows: (1) Length in nucleotides for each contig; (2) Coding Sequences (CDS) in the Forward Strand (light blue); (3) CDS in the reverse strand (dark blue); (4) Strain Specific Genes (SSGs) in the forward strand (light purple); (5) SSGs in the reverse strand (dark purple); (6) GC Skew (gray). Skew and gene distribution follow that of a typical Firmicute genome. The Strain Specific Genes in the contigs that belong to the chromosome appear to be randomly distributed, whilst they seem to be concentrated in the contigs 12 and 13, which are probably the ones belonging to megaplasmids. The circular plot was done with Circos software (*Krzywinski et al., 2009*).

**The *Exiguobacterium* strain N139 belongs to the *chiriqhucha* species along with *E. pavilionensis* str. RW-2 and *Exiguobacterium* sp. GIC31**

A phylogenomic reconstruction (Fig. 1A) placed the strain N139 as most similar to *Exiguobacterium* str. GIC31 (*Vishnivetskaya et al., 2014*) as well as to *E. pavilionensis* str. RW-2 (*White III, Grassa & Suttle, 2013*). ANI and AAI calculations of all clade II *Exiguobacterium* strains were performed and compared to our N139 strain, suggesting that *E. pavilionensis* str. RW-2, *Exiguobacterium* sp. GIC31 and this N139 strain belong to the same species since they share ANI values above 97% (Table S2) (*Goris et al., 2007*).
Typically, the ANI values between genomes of the same species are above 95% (e.g., *E. coli*). ANI and AAI scores of all pairwise comparisons of the three proposed *Exiguobacterium chiriqhucha* strains exceed the 97% threshold (data not shown). Also, relying on the ANI and AAI calculations, it was concluded that the outgroup of the *E. chiriqhucha* species could be *E. mexicanum*.

### *Exiguobacterium* clade II pangenomes

To further understand the genomic properties of *E. chiriqhucha* str. N139 and its taxonomic positioning, we built the pangenome of nine *Exiguobacterium* strains from clade II, whose complete genomes were available at the time of analysis. This pangenome is composed of 5,267 genes; 2,116 of them belonging to the core genome and 1,664 being Strain Specific Genes (SSGs). The resulting pangenome shows a very conserved and cohesive pool of genes, despite their evolutionary distance and their remote geographic locations. Over two thousand genes compose the core genome, which represents a large core genome when compared to other pangenomes, and taking into account that the average genome size of *Exiguobacterium* strains, which is approximately three thousand genes. The SSGs are represented in a heatmap on Fig. S3 where the clusterization of the *Exiguobacterium* strains is based on the presence (and abundance) or absence of their COG assignation. *Exiguobacterium* sp. S17 and *E. mexicanum* are exceptional for the fact that they possess a large pool of SSGs (Table S3). We speculate that some of these SSGs could have been acquired by Horizontal Gene Transfer (HGT) and retained to adapt to these diverse environments, or equally likely, lost in some of the living taxa, due to lack of selective pressure in their respective niches.

Fifty-nine of the SSGs found in the *E. chiriqhucha* str. N139 were mapped on its genome to see if their distribution followed some bias (Fig. 3). Throughout the contigs that are putative chromosomal regions, the SSGs appear to be randomly distributed. However, some of the SSGs are concentrated in the contigs 12 and 13, supporting the previous idea that these contigs may be part of the megaplasmids seen in the PFGE analysis (Fig. S1). COGs were assigned to the SSGs as previously described for the *E. chiriqhucha* str. N139 genome. For the whole set of SSGs of the pangenome, COGs were successfully assigned to 66% of the genes, and are represented in a heatmap (Fig. S3). However, most of the N139 SSGs could not be assigned to COGs, and for those that were successfully classified, the vast majority falls in the S and R (Poorly Characterized) COG categories, leaving open questions on which may be the unique strategies that N139 employs to adapt to the particular environment of Laguna Negra.

### Main metabolic pathways, amino acids, nucleotides and cofactors

Based on its genomic content, *E. chiriqhucha* str. N139 is probably a chemoheterotroph since it has two copies of *aioB* arsenite oxidase, which means it could obtain energy from arsenite oxidation. It has the complete pathway for glycolysis and it could synthesize acetyl-CoA, succinyl-CoA and isobutanoyl-CoA. It is a heterolactic fermenter, being able to produce lactate from pyruvate and ethanol from acetaldehyde. It has a complete TCA cycle, and it lacks the first two steps of the pentose phosphate pathway, but the rest of the

pathway is present. Hence, its central metabolism is similar to *B. subtilis* (*Blencke et al., 2003*; *Blencke et al., 2006*), but *E. chiriqhucha* str. N139 can synthesize more fermentation products, namely ethanol and formate. *E. chiriqhucha* str. N139 lacks the routes for synthesizing *de novo* phenylalanine and tyrosine, as well as the Branched Chain Amino Acids (BCAA). However, it can synthesize tyrosine from phenylalanine, since it has the phenylalanine-4-hydroxylase regulated by a Tyr (UAC codon) T box riboswitch. Despite lacking the complete pathways for BCAA biosynthesis, it preserves the *ilvE* gene, a BCAA aminotransferase, which could probably synthesize any of the three BCAAs from available precursors. An interesting note on its tryptophan biosynthesis is that its biosynthetic operon is split in two transcription units: *trpEG* and *trpDCFBA*, which are separated in the chromosome, but co-regulated by a Trp T box riboswitch. Although this regulation is common in Firmicutes (*Gutierrez-Preciado et al., 2005*; *Gutiérrez-Preciado, Yanofsky & Merino, 2007*), the genome context of the *trp* operon is not, and it is interesting that this separation takes place at the synthesis of anthranilate. Moreover, the *trpEG* genes are regulated by a single T box, whilst the *trpDCFBA* operon is regulated by two T boxes in tandem. This could either mean that the separation of the pathway is a recent event and the regulation is being settled in order to coordinate both transcriptional units; or that this strain requires anthranilate (the product of *trpEG*) for something else. Certainly, one possibility is that *E. chiriqhucha* str. N139 exports anthranilate for a synthrophy with a partner(s) and the subsequent steps of the tryptophan biosynthetic pathway require a stricter regulation in order for the genes *trpDCFBA* to be expressed. Since *E. chiriqhucha* str. N139 lacks the biosynthetic pathways for five amino acids, a likely scenario is that this bacterium is sharing metabolites with other partners in Laguna Negra. This is supported by the observation that it is able to form part of a biofilm, and that in all of the amino acids tested it can only grow on serine and asparagine as a sole carbon source (see Table S1). Based on the metabolite tracer from Pathway Tools, it can be inferred that *E. chiriqhucha* str. N139 could synthesize phenylalanine as well as valine from serine or asparagine. In the same fashion, it cannot grow with phenylalanine as the sole carbon source. Therefore, the configuration of the genes involved in amino acid metabolism might represent a requirement of amino acid syntrophy that needs further exploration and testing. A second possibility is that *E. chiriqhucha* str. N139 is utilizing anthranilate for some novel pathway. Anthranilate cannot be in excess with respect to tryptophan, since its excess could decrease the availability of phosphoribosyl pyrophosphate (PRPP) for histidine synthesis (and other reactions) (*Merino, Jensen & Yanofsky, 2008*). This novel pathway could be involved in different functions that require either tryptophan or anthranilate as intermediates. Examples of these functions are quorum sensing molecules in *Pseudomonas aeruginosa* (*Farrow & Pesci, 2007*), plant hormones in *Azospirillum brasilense* (*Ge, Xie & Chen, 2006*), violacein in *Chromobacterium violacein* (*Antônio & Creczynski-Pasa, 2004*) or antibiotics as in *Streptomyces coelicolor* (*Amir-Heidari, Thirlway & Micklefield, 2008*).

The regulation of biosynthetic and transporter genes through riboswitches is common in Firmicutes, specially the members of Bacilli class. It has also been observed that transport and biosynthesis of the same metabolite tend to be part of a regulon mediated by *in cis* elements, like riboswitches (*Gutiérrez-Preciado et al., 2009*). Methionine can be synthesized

and imported through several strategies. Several SAM riboswitch regulated operons coding for Met transporters were identified in the genome of *E. chiriqhucha* str. N139, as well as canonical *met* biosynthetic genes. An interesting case is the methionine salvage pathway, whose genes are encoded in two divergent operons, both regulated by divergent SAM riboswitches. Both operons must be transcribed in order for the Yang cycle to be completed. In one operon, genes *mtnK* and *mtnA* are transcribed along with three ribose transporters, *rbsB, rbsC* and *araG*. Lysine biosynthesis (from aspartate via diaminopimelate) and transport are part of a regulon under the lysine riboswitch. Furthermore, through the identification of riboswitches, two transporters from the NhaC family were annotated: one as a methionine transporter (SAM riboswitch), and the other one as a lysine transporter (LYS riboswitch). This strategy of improving gene annotation through the knowledge of the gene's regulation has been previously explored (*Rollins, 2002*; *Gutiérrez-Preciado & Merino, 2012*; *Gutiérrez-Preciado et al., 2015*).

### Cofactors

Thiamine can be synthesized *de novo*, its biosynthesis and its uptake are regulated by the TPP riboswitch. Moreover, the analysis of *E. chiriqhucha* str. N139 genome indicates that a new thiamine transporter could be present in this bacterium. The gene exiN139_02072 is automatically identified as a membrane protein, but it seems to be regulated by a TPP riboswitch. Experimental evidence is needed for the confirmation and characterization of this transporter, which could unveil a new family of thiamine transporters. Riboflavin biosynthesis and transport (RibU) are also co-regulated through a FMN riboswitch.

### Nucleotides

In the genome of *E. chiriqhucha* str. N139 the purine *de novo* biosynthetic pathway is encoded in a huge transcription unit regulated by a purine riboswitch. Other transcription units in the same regulon include a monocystronic GMP synthase, and genes involved in adenine and adenosine salvage pathway.

## GENOMIC ADAPTATIONS TO AN EXTREME ENVIRONMENT

Laguna Negra is an aquatic ecosystem that harbors extreme environmental conditions such as high levels of UV-B (10.65 wm2), high salinity levels (32%), scarce nutrients, particularly phosphorous (<005 mg/l), high metal contents including the metalloid arsenic (3 mg/l), an alkaline pH and large daily temperature fluctuations (ranging from 20 °C during the day to −40 °C at night) (*Flores et al., 2009*); (see Table 1).

### Resistance to metals and metalloids

In Laguna Negra, ubiquitous Arsenic enters the *E. chiriqhucha* str. N139 cells through existing transporters due to its high structural similarity with other molecules (*Rosen, 1999*) and induces oxidative stress responses (*Oremland & Stolz , 2003*). Furthermore, arsenite (AsO$_2$H) and arsenate (AsO$_4^{3-}$), are both toxic molecules. Arsenite binds to reduced cysteines in proteins inactivating them, and arsenate is a molecular analog of phosphate and therefore inhibits oxidative phosphorylation (*Oremland & Stolz , 2003*). Arsenate is far less toxic than arsenite, hence the oxidation of arsenite is considered a

detoxification process. However, the oxidation of arsenite to arsenate, when coupled to the reduction of oxygen to water, is an exergonic process, and it has been suggested that at least some bacteria may derive energy out of this process (*vanden Hoven & Santini, 2004*). *E. chiriqhucha* str. N139 has an arsenite oxidase, AioB, enabling it to oxidize arsenite. This is an important metabolic capability, because it uses arsenite as an electron donor. Moreover, from a bioremediation point of view, this former metabolic feature is important since arsenite is more soluble than arsenate, so it can facilitate the removal of As in solution. *E. chiriqhucha* str. N139 also has an arsenite efflux pump, ArsB, as well as an ATPase that provides energy to ArsB for extrusion of arsenite and antimonite, ArsA, co-transcribed with ArsD, an arsenic chaperone for the ArsAB pump (*Páez-Espino et al., 2009*). Hence, this bacterium can probably detoxify and extrude As, as well as oxidize arsenite acquiring energy from this process. These ArsAB and ArsD proteins are also present in *Salinivibrio* strains isolated from the Laguna Socompa. However, these *Salinivibrio* strains also have ArsC, a cytoplasmic oxidoreductase that reduces arsenate to arsenite in a ATP-glutathione-glutaredoxin dependent way (*Gorriti et al., 2014*). *E. chiriqhucha* str. N139 lacks significant homologs to this gene as well as significant homologs to *B. subtilis' arsC* gene.

*E. chiriqhucha* str. N139 also possesses redundancy for mercury detoxification, harboring four paralogous copies of *merA*. Briefly, MerA is the key detoxification enzyme of the mercury resistance system, reducing $Hg^{2+}$ to $Hg^0$ (*Silver & Phung, 2005*). Hg is toxic due to its high affinity to sulfur (*Nies, 2003*) and usually, *mer* resistance genes are co-transcribed in an operon whose dissemination is common by horizontal gene transfer (HGT) (*Barkay, Miller & Summers, 2003*). In this organism, two copies of *merA* are present in a monocystronic fashion; a third one is transcribed with a hypothetical protein. A fourth copy is co-transcribed with *merR*, the regulatory protein of the system. Two *mer* transporters which uptake Hg and *merP*, a transporter with a Sec-type signal, which could import Hg as a neutral chloride or hydroxide and deliver it to the other Mer transporters, which will finally transfer it to MerA.

The most common mechanism of resistance to metals consists of efflux pumps for inorganic ions. However, As and Hg resistance mechanisms are unique in the sense that these elements are reduced to lower their toxicity (*Silver & Phung, 2005*), instead of being exported. *E. chiriqhucha* str. N139 is resistant to cadmium, zinc, cobalt, and copper by pumping it out from the cell. It has two membrane embedded $Cd^{2+}$ efflux pumps, one of which can also extrude zinc and cobalt; two paralogous copies of *copA* and *copB*, two P-type ATPase systems for exporting copper, and *cueR*, a sensing cytoplasmic Cu that protects periplasmic proteins from copper-induced toxicity (*Orell et al., 2010*). *copB* is transcribed monocystronically, and each of the *copA* genes form an operon co-transcribed with a copy of *copZ*, a copper chaperone, but one is co-transcribed with a glutaredoxin, whilst the other is co-transcribed with *csoR*, a copper-sensitive operon repressor.

Additionally, this microorganism lives in a low phosphorous environment, and relies on strategies for phosphorous uptake, like the presence of high-affinity Pi transporters and its regulation (*pstS*, *pstCAB*, *phoB*, *phoR*, *dedA* and *ptrA*) and genes for polyphosphate storage and breakdown (*ppk* and *ppx*). Organisms that scavenge phosphate can sometimes uptake the structurally similar arsenate ion, and hence also depend on arsenate detoxification

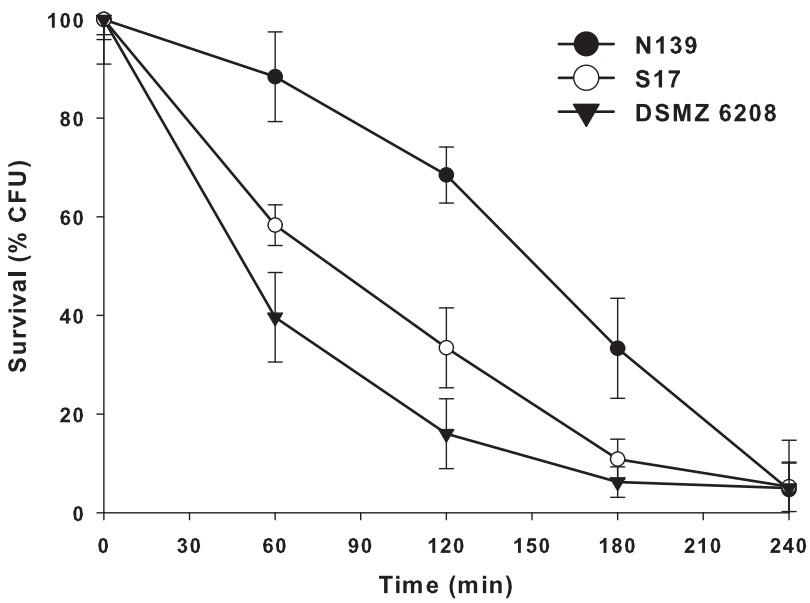

**Figure 4** **Effect of ultraviolet B (UV-B) radiation on *Exiguobacterium* strains.** Percentage survival to UV-B radiation of str. N139 (dark circle), str. S17 (light circle) and str. DSMZ 6208 (dark triangle). The influence of UV-B radiation was studied by exposing liquid cultures to increasing doses, varying exposure times between 0 and 240 min.

mechanisms. It is also able to thrive in the alkaline environment of Laguna Negra since its genome code for all the typical antiporters present in alkaliphilic bacilli (*nhaC, nhaP,* the *mpr* operon, *yhaU, norM* and *mleN*). These antiporters present also contribute to a moderate salinity resistance this could also be related with the maintenance of metal resistance strategies in its genome, since it has been shown that lowering the salinity can lead to enhanced sensitivity to cadmium, cobalt and copper.

## Resistance to UV radiation

Another extreme environmental condition in Laguna Negra is high ultraviolet radiation, particularly UV-B (*Flores et al., 2009*; *Ordoñez et al., 2009*). In order to determine if *E. chiriqhucha* str. N139 can cope with this constant stress, we measure the effect of colony survival of different *Exiguobacterium* strains exposed to UV-B radiation (Fig. 4). More than 25% of the colonies survive after 3 hrs of constant UV-B radiation. This contrasts with the other *Exiguobacterium* strains which rapidly start to decay, even though one of them, str. S17 was isolated from a neighbor lake, Laguna Socompa, in the HAALs (*Ordoñez et al., 2013*).

Bacteria have different UV damage repair pathways, including photoenzymatic repair (PER), nucleotide excision repair (NER) also called dark repair, and recombinational repair (PRR) or post-replication repair (*Goosen & Moolenaar, 2008*). *E. chiriqhucha* str. N139 has three genes (exiN139_00335 (*phrB*), exiN139_01768 and exiN139_00235) related to photolyases, which are involved in PER. They use UV as energy source (using FADH and transferring electrons) and catalyze the monomerization of cyclobutyl pyrimidine dimers. The gene exiN139_00335 only has homologues in Firmicutes including other known *Exiguobacterium,* and exiN139_ 01768 has homologues in Firmicutes, Cyanobacteria,

α- and γ- Proteobacteria, and Euryarchaeotes. Additionally, exiN139_00235 is a cryptochrome, which are flavoproteins related to photolyases. Cryptochromes do not repair DNA and are presumed to act in other (unknown) processes, such as entraining circadian rhythms (*Yuan et al., 2012*). It is worth to remark that from these proteins only exiN139_01768, annotated as Deoxyribodipyrimidine photo-lyase-related protein, contains significant homologs in the genomes of the two *Exiguobacterium* compared in the UV-B radiation resistant assay (Fig. 4).

E. chiriqhucha str. N139 has also genes for NER. Its genome encodes the UvrABC endonuclease, a complex that recognizes DNA damage, binds to the damaged segment and cleaves it. Additionally, it codes for PcrA (also known as UvrD), a helicase in charge of removing the excised segment recognized and cleaved by UvrABC. These genes are regulated by the SOS response, which uses LexA as a repressor inactivated by RecA (*Minko et al., 2001*). E. aurantiacum and E. str. S17, contain homologs for the *uvrB/uvrC* gene, and the *pcrA* and *recA* regulatory genes.

Regarding the PRR, *E. chiriqhucha* str. N139 encodes for RecA, which recognizes SSB and cleaves UmuD, which becomes UmuD' and binds UmuC to generate polymerase V, which in turn repairs damages, sometimes causing mutations. The gene exiN139_03003 may produce polymerase IV that is also involved in DNA damage repair (*Sommer et al., 1998*). UmuC is found in the genome, however UmuD is missing. It is possible that a protein highly similar to an existing copy of LexA may be taking its role, given that both can be cleaved by RecA and are present in *Exiguobacterium*. UmuD and UmuC are not present in the genomic sequences of *E. aurantiacum* and *E.* str. S17.

E. chiriqhucha str. N139 appears robust towards UV-B radiation, possessing several mechanism to cope with this constant stress. Moreover, the two strains proposed to be part of the same *chiriqhucha* species, *E. pavilionensis* str. RW-2 and *E.* str. GIC31, contain the same set of genes described above to cope with UV-B radiation, another unifying property of these strains. The strains RW-2 and N139 contain two homologs for Deoxyribodipyrimidine photo-lyases, while GIC31 contains three.

## Living in syntrophy?

Finally, it is likely that *E. chiriqhucha* str. N139 participates in biofilm formation in Laguna Negra along with other bacteria (unpublished results). Analyses of other *Exiguobacterium* have shown that they participate in marine biofilms interacting with other Firmicutes and Proteobacteria (*López et al., 2006*; *Carneiro et al., 2012*). Evidence of possible biofilms associated genes originates from two *loci* present in the genome of *E. chiriqhucha* str. N139. The first *locus* encoding a protein capable of producing alginate, a linear co-polymer of two uronic acids that is produced in its acetylated form by some bacteria for adherence of these bacteria to target cell walls by the creation of a biofilm (*Ramphall & Pier, 1985*). The second *locus* codes for the arginine deiminase system, which can function at very low pH and is thought to be a critical factor in oral biofilm pH homeostasis (*Burne & Marquis, 2000*).

## CONCLUSIONS

*E. chiriqhucha* str. N139 lives in a high-altitude, salted lake exposed to intense UV radiation, about 300 km away from the nearest ocean, the Pacific. Many factors in *E. chiriqhucha* str. N139 metabolism, such as the its needs to uptake certain intermediates like phenylalanine and BCAAs, and the possible excretion of the overproduced anthranilate, suggest that it is a key player in the amino acid metabolism of a microbial consortium that inhabit Laguna Negra. Moreover, the excess of anthranilate that it may produce could be directed to some novel pathway that remains to be uncovered, such as a new antibiotic, a new pigment or a new quorum sensing molecule.

The genome of *E. chiriqhucha* str. N139 contains all the necessary strategies to cope with all the environmental stresses that simultaneously co-occur in Laguna Negra. This *Exiguobacterium* is able to detoxify metals like arsenic, mercury, cadmium, zinc, cobalt, and copper; it has a complete defense system against UV damage; and it is also able to thrive in the alkaline environment of Laguna Negra. (*Ventosa, Nieto & Oren, 1998*). With all these characteristics, *E. chiriqhucha* str. N139 is an excellent candidate for future biotechnological research.

Although our study generates more questions than the ones it could solve, by sequencing its genome we have gained insights on the strategies the strain N139 employs for thriving in its habitat. From its Strain Specific set of genes, only 23 out of 59 could be annotated and classified to a COG, and still, most of the COG-classified genes belong to the poorly characterized category. This set of genes of unknown function require further experimental work to completely unveil how the strain N139 is adapting to the extreme environment of Laguna Negra.

## DESCRIPTION OF *EXIGUOBACTERIUM CHIRIQHUCHA SP. NOV.*

*Exiguobacterium chiriqhucha* (chi.ri.qhu.cha. (/ʃi ri ku tʃa/)) Quechua. Adj. *chiri*: cold, freezing; Quechua. Noun. *qhucha:* lake, pond. *chiriqhucha* of or belonging to a cold lake, referring to the common habitat of these three species). Members of the species *Exiguobacterium chiriqhucha*; inhabitants of freshwater ponds, saline ponds; distinguishable by their 16S rRNA sequences; accession numbers are: JMEH00000000 for the str. N139 genome, ATCL00000000 for the RW-2 genome (*White III, Grassa & Suttle, 2013*) and JNIP00000000 for the GIC31 genome (*Vishnivetskaya et al., 2014*). The three strains that so far comprise this species form orange shiny colonies and are Gram-positive, rod-shaped, facultative anaerobes and motile via peritrichous flagella (*Miteva, Sheridan & Brenchley, 2004*; *Vishnivetskaya, Kathariou & Tiedje, 2009*; *White III, Grassa & Suttle, 2013*). Two of them, str. RW-2 and str. GIC31 were isolated from permanently cold environments (Pavilion Lake and Glacier Ice, Greenland; (*White III, Grassa & Suttle, 2013*; *Vishnivetskaya et al., 2014*)) whilst the str. N139 was isolated from Laguna Negra, a HAAL which temperature can drop to −30 °C. Their temperature range of growth is from a minimum (str. GIC31) of 2 °C to a maximum (str. RW-2) of 50 °C; the pH range for growth is from 5 to 11 in str. RW-2 and from 7 to 9 in str. N139 (*Miteva, Sheridan &*

*Brenchley, 2004*; *White III, Grassa & Suttle, 2013*). The three strains possess cold-shock proteins and have a G + C content of 52% (*White III, Grassa & Suttle, 2013*; *Vishnivetskaya et al., 2014*). The type strain is RW-2.

**Abbreviations**

| | |
|---|---|
| **HAALs** | high altitude Andean Lakes |
| **ANI** | Average Nucleotide Identity |
| **AAI** | Average Amino Acid Identity |
| **LM** | Lake Medium |
| **SSGs** | Strain Specific Genes |
| **HGT** | Horizontal Gene Transfer |
| **BCAA** | Branched Chain Amino Acids |
| **PER** | Photoenzymatic Repair |
| **NER** | Nucleotide Excision Repair |
| **PRR** | Post Replication Repair |

## ACKNOWLEDGEMENTS

We thank Jerome Verleyen for computer support and the Instituto de Biotecnología-UNAM for allowing us access to its computer cluster. We would also like to thank Eric Nawrocki for his guidance with the *ssu-align* program.

### Funding

AGP was supported by SECITI Postdoctoral Fellowship 029/2013, and at present she is a recipient of a junior postdoctoral contract in the Prometeo Program from the Gobierno Valenciano; CVC by a fellowship from CONACyT Mexico (462083); MRP by the EU Marie Curie Initial Training Network (ITN) Symbiomics: Molecular ecology and evolution of bacterial symbionts [FP7-PEOPLE-2010-ITN]; OFO is a recipient of a Postdoctoral Fellowship from Bunge and Born Foundation; TRP is funded by a Postdoctoral Fellowship from CONACyT Mexico; JVA is supported by a fellowship from CONACyT Mexico (207187); EAR is funded by a NSF grant 1136602. This work was also supported by BFU2015-64322-C2-1-R (co-financed by FEDER funds and Ministerio de Economía y Competitividad, Spain) and PrometeoII/2014/065 (Conselleria d'Educació, Generalitat Valenciana, Spain) to AL and AM, respectively; and by a grant from WWF-Alianza Carlos Slim to VS. There was no additional external funding received for this study. The funders had no role in study design, data collection and analysis, decision to publish, or preparation of the manuscript.

### Grant Disclosures

The following grant information was disclosed by the authors:
SECITI Postdoctoral Fellowship: 029/2013.
Gobierno Valenciano.

CONACyT Mexico: 462083, 207187.

EU Marie Curie Initial Training Network (ITN) Symbiomics: Molecular ecology and evolution of bacterial symbionts: FP7-PEOPLE-2010-ITN.

Bunge and Born Foundation.

CONACyT Mexico.

NSF: 1136602.

co-financed by FEDER funds and Ministerio de Economía y Competitividad, Spain: BFU2015-64322-C2-1-R.

Conselleria d'Educació, Generalitat Valenciana, Spain: PrometeoII/2014/065.

WWF-Alianza Carlos Slim.

## Competing Interests

Valeria Souza is an Academic Editor for PeerJ.

## Author Contributions

- Ana Gutiérrez-Preciado conceived and designed the experiments, analyzed the data, wrote the paper, prepared figures and/or tables, reviewed drafts of the paper.
- Carlos Vargas-Chávez conceived and designed the experiments, analyzed the data, prepared figures and/or tables, reviewed drafts of the paper.
- Mariana Reyes-Prieto analyzed the data, prepared figures and/or tables, reviewed drafts of the paper.
- Omar F. Ordoñez performed the experiments.
- Diego Santos-García analyzed the data, prepared figures and/or tables.
- Tania Rosas-Pérez analyzed the data.
- Jorge Valdivia-Anistro performed the experiments, prepared figures and/or tables.
- Eria A. Rebollar conceived and designed the experiments, performed the experiments, analyzed the data, wrote the paper, reviewed drafts of the paper.
- Andrés Saralegui and María Eugenia Farías performed the experiments, contributed reagents/materials/analysis tools.
- Andrés Moya analyzed the data, reviewed drafts of the paper.
- Enrique Merino analyzed the data, contributed reagents/materials/analysis tools, prepared figures and/or tables, reviewed drafts of the paper.
- Amparo Latorre and Valeria Souza conceived and designed the experiments, analyzed the data, wrote the paper, reviewed drafts of the paper.

## DNA Deposition

The following information was supplied regarding the deposition of DNA sequences:

This Whole Genome Shotgun project has been deposited at GenBank under the accession JMEH00000000. The version described in this paper is version JMEH001000000.1; BioSample SAMN02732272.

## Data Availability

The raw data is in the genome fasqual and sff files available here: http://www.ibt.unam.mx/biocomputo/downloads/rawGenome.tar.gz.

## Supplemental Information

Supplemental information for this article can be found online at http://dx.doi.org/10.7717/peerj.3162#supplemental-information.

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
