# Peer review of "The genomic sequence of Exiguobacterium chiriqhucha str. N139 reveals a species that thrives in cold waters and extreme environmental conditions"

_PeerJ, doi:10.7717/peerj.3162_

## Round 0.1 · original submission · Major Revisions

Ana Gutiérrez-Preciado et al. present a genomic analyses of strain N139 isolated from an Andean Lake. This study merits publication only after a complete revision. It appears as a working draft for the moment. In addition to the two referees remarks (If you have any problem understanding referees remarks let me know), I have a number of remarks in order to help improve this text prior publication.

First, the name Exiguobacterium pavilionensis is not a formally proposed name (see e.g. http://www.bacterio.net/exiguobacterium.html) (Line 226). Apparently this species name was coined by White et al. 2013 (PMCID: PMC3738901). I suggest the authors perform a detailed genome-based taxonomic analyses and propose a formal species name to encompass strain N139 and related strains (e.g. RW-2). Some useful data is already available (e.g. Table S1) Butiric acid, Serine, and Tween 40 are positive in N139 and negative in E. mexicanum. On the other hand Cellobiose, Mannitol and Amydalign are positive for E. mexicanum and negative for N139. In silico DNA-DNA distance, and AAI need to be calculated among all available (see your clade II) type strains e.g. E. mexicanum. N139 is probably a new species.
Second, phenotypes are mentioned concerning UV and metal resistance, but data to prove these phenotypes was omitted from the text (e.g. Lines 31, 73, 104).

Third, I had the impression that the presence of the gene meant the presence of the protein (e.g. transporters) Line 344 - 418. Revise and make sure to refer to the gene 'as coding a protein...'. Also this part is very descriptive and even lacks a comparison with other genome sequences obtained from bacteria isolated from Andean Lakes e.g. salinivibrios PMCID: PMC4094778. This can be useful e.g. to determine if gene dosage is higher/lower in N139 for the different genetic mechanisms to coupe with UV and metals. How N139 genome compares with other genomes from the Andean Lakes in terms of genes related to adaptation?

Fourth, The text needs to be enhanced and refined. The title needs to be re-phrased; It does not reflect the contents of the text.

Fifth. I am not satisfied with the genome sequence quality, but will accept it is generated by 454. Please make sure to respond to all ref#3 remarks.

Minor.
Lines 100-103. Remove this part.
Line 341. What do you mean by 005 mg/l
Line 431. Did the authors see phosphate granules in the cell?
Line 443-448. Remove this paragraph. Useless.

Reviewer 1 ·

Basic reporting

I would like to congratulate the author for the manuscript. A lot of work was performed in order to explore the genome of the Exiguobacterium pavilionensis str. N139, and its ability to deal with UV-radiation and metal resistance. However some point need to be clarified.

Experimental design

Lines 122-127: Phylogenetics analysis: why only 16S gene sequence was used in the reconstruction if the complete genome sequences from these organisms are available?
How ANI was calculated? What parameters of coverage and identity were used? Furthermore in this section some results can be found (Lines: 146-151)
What are the parameters used in Newbler and MIRA assembler?
What is the phred quality cut-off used in the sequences?
The lines 160-161 should be moved to results section.
All COGs clusters were used? What was the COG version used? What was the rationally to run this COG analysis? Please clarify these points.
What is the N50 of the assembly?
The category R and S of COG do not have a function assigned, are only general prediction only, but this categories were considered as annotated, why?

Validity of the findings

How the plasmids were identified?
Why only ANI was evaluated? AAI and Karling signature shows the same panorama? I suggest include these analysis to support, or not, this result.
Lines 246-247: How is possible compare a core genome with pangenomes, please clarify.
Lines 249-250: This statement is speculative. Any HGT analysis was performed? Perhaps these genes could have a high mutation rate?
Lines 327-329: Please clarify, how a new possible thiamine transporter was identified?
Line 375: Please fix the citation “[66]”??
The result and discussion section are very descriptive and lacks references to support the findings.
Conclusions
The conclusion can be shortened and some clauses can be moved discussions.
It is intriguing, in table 4, that COG category A was not mapped in this organism. Why?

Additional comments

I have no more questions and would like to congratulate the authors for the efforts

Reviewer 2 ·

Basic reporting

The manuscript investigates how the bacteria Exiguobacterium pavilionensis isolated from High Altitude Andean Lakes could be adapt to poor nutrient availability and the ability of the strain survive with high UV-B radiation and metal resistance. The authors describes with details the characteristic genetics of Exiguobacterium pavilionensis str. N139. The main analysis are appropriate in my opinion, but there is some aspects of taxonomy position and genomic analysis could be improved (see Experimental Design). The data clearly demonstrate that there is evidences that Exiguobacterium pavilionensis is able to adapt in hostil environments.

Experimental design

The Material and Methods section are well written, however I suggest to revise this section. There is some parts in the Result and Discussion section that should be present in this section. In addition, the figures are not visible in the supplemental material file. Please, check the file.

Additional comments for authors:

Line 131. Why the sequences from Exiguobacterium did not downloaded from SILVA database as other sequences?

Line 153-151. Did you de novo or use some reference to assembly of genomes? Is not clear in the text.

Validity of the findings

The Results and Discussion covers the aim of the manuscript, however there is some aspects that should be improved. Please, check the topics that contains sentences of the Material and Methods section.

Follow, specific comments:

Line 190-195. The sentences should be in Material and Methods.

Line 201. "This strain". The font word is different from the text.

Line 203. The regions hr-A1, hr-AB and hrA2 are adjacently? Is a unique region? Is there any similarity of Exoguibacterium genomes?

Line 209. Does hr-AB-B = hr-AB ??

Line 214. What genes are annotated in contig 14? I think that is interesting to cite the predict genes of the contig. Also, I suggest the authors use some tools to predict genomic islands. The results could be clarified the origin of contig.

Line 227. I think only ANI and 16S rRNA are not enough to provide a secure taxonomic position. I suggest additional genomic taxonomic analysis, like hybridization in silico and/or AAI, MLSA.

Line 232. The sentence "..since its genome is 400 genes bigger than the remaining members of the E. pavilionensis clade." is not very informative. You can't to to confirm this sentence based on number of genes.

Line 236-244. I did not see any reason for two different analysis for pangenome( first, 4 genomes; second, 9 genomes).

Line 237. I think the term "outgroup" is not make sense for pangenome analysis.

Line 242. What genomes of Clade II were sequenced and included in the pan genome analysis? Also, I think that its important the authors to cite the geographic locations from the strains of pangenome.

Line 246. In the text, is not clear the term Single Copy Core. What's the difference of core genome? Are you removed all paralogs genes of the analysis? I suggest explain with more detail this analysis in Material and Methods.

---

## Round 0.2 · Minor Revisions

Please make sure to correct your paper according to the remarks of referee #1.

Reviewer 1 ·

Basic reporting

I would like to congratulate the author for the manuscript. A lot of work was performed in order to explore the genome However some point yet need to be clarified.
Line 37: “cosmopolita or cosmopolitan”?
Lines 85-87: There is any reference supporting this phrase, or the statement is “common sense”?

Experimental design

Lines 140-146: This reviewer suggests authors to write the parameters used in pre-processing (i.e. Phred quality) and assemble process in the manuscript
Lines 146-148: “The final assembly 147 consists of 23 contigs, the smallest one being 457 bases in length and the largest 1.5 Mb, with an average coverage of 85×.” Should be moved to results section.
What is the e-value cut-off used in HMMER search? Please add the information in manuscript.
Please add in the manuscript the parameters used in ANI and AAI calculations.
Line 184: “aurantiacum, from a potato was, in the UK”, isolated?

Validity of the findings

Line 211: “Genome properties” Please add the N50 information in the manuscript, or in Table 3.
Lines 237-241: “It was concluded that the sequences present in the plasmids are shared within different Exiguobacterium strains, displaying a highly dynamic behavior, and as a result, determining which of our contigs correspond to the three megaplasmids observed in the PFGE experiments was not possible (see Supplemental Material).” Please re-write the phrase, it is confuse. What was not possible?
Line 302: Please fix the reference in manuscript “[48–50]”, and write de correct reference.
Lines 324-329: What are the genomics adaptations to an Extreme Environment? The authors only list the environment conditions, and in Table 1 list some bacterial features.
Lines 414-423: This new sections is already know, there are others manuscript showing a similar scenario (where are the references?) Or this mechanism is new and described for the first time?

If the authors are proposing that these 3 isolates (RW-2; GIC31 and N139) belongs to a new (and same) species (Exiguobacterium chiriqhucha), where is the description of the species?

Figure legends; lines 538-539: “Figure 1. Evolutionary history of the genus Exiguobacterium. A) Phylogenetic reconstruction with the 16S rRNA gene using MrBayes (see Methods)”. The three was constructed using a phylogenomic approach (as described in lines 134 – 135) or 16S gene (as described in figure legend)?

Additional comments

It is remarkable the improve in manuscript, however before acceptance a few adjusts are necessary.

---

## Round 0.3 · accepted · Accept

Ana congratulations on the accepted MS.